# Generality and opponency of rostromedial tegmental (RMTg) roles in valence processing

**Hao Li, Dominika Pullmann, Jennifer Y Cho, Maya Eid, Thomas C Jhou***

Department of Neuroscience, Medical University of South Carolina, Charleston, United States

**Abstract** The rostromedial tegmental nucleus (RMTg), a GABAergic afferent to midbrain dopamine (DA) neurons, has been hypothesized to be broadly activated by aversive stimuli. However, this encoding pattern has only been demonstrated for a limited number of stimuli, and the RMTg influence on ventral tegmental (VTA) responses to aversive stimuli is untested. Here, we found that RMTg neurons are broadly excited by aversive stimuli of different sensory modalities and inhibited by reward-related stimuli. These stimuli include visual, auditory, somatosensory and chemical aversive stimuli, as well as "opponent" motivational states induced by removal of sustained rewarding or aversive stimuli. These patterns are consistent with broad encoding of negative valence in a subset of RMTg neurons. We further found that valence-encoding RMTg neurons preferentially project to the DA-rich VTA versus other targets, and excitotoxic RMTg lesions greatly reduce aversive stimulus-induced inhibitions in VTA neurons, particularly putative DA neurons, while also impairing conditioned place aversion to multiple aversive stimuli. Together, our findings indicate a broad RMTg role in encoding aversion and driving VTA responses and behavior.
DOI: https://doi.org/10.7554/eLife.41542.001

*For correspondence:
jhou@musc.edu

Competing interests: The authors declare that no competing interests exist.

## Introduction

The rostromedial tegmental nucleus (RMTg), also called the tail of the ventral tegmental area (tVTA), is a GABAergic nucleus first described in 2009 as a major inhibitory input to neurons in the ventral tegmental area (VTA) (*Jhou et al., 2009*; *Kaufling et al., 2009*). Previous studies have shown that RMTg axons overwhelmingly synapse on tyrosine hydroxylase (TH) positive neurons in the VTA and substantia nigra, and electrical stimulation of the RMTg dramatically suppresses DA neuron firing (*Balcita-Pedicino et al., 2011*; *Bourdy et al., 2014*). Subsequent studies have further suggested an important RMTg role in conveying information about aversive stimuli onto VTA and substantia nigra DA neurons and mediating behavioral responses to these aversive stimuli (*Brown et al., 2017*; *Hong et al., 2011*; *Jhou et al., 2009*; *Jhou et al., 2013*; *Vento et al., 2017*). However, some fundamental questions about the proposed RMTg role in aversive encoding have not been addressed.

Most prior studies had only tested RMTg firing responses to a limited range of aversive stimuli, for example footshocks, airpuffs, and their predictive cues (*Hong et al., 2011*; *Jhou et al., 2009*). One notable exception is a recent study that examined 11 distinct aversive stimuli, but found that only 3 of them increased RMTg expression of the immediate-early gene c-Fos (*Sánchez-Catalán et al., 2017*). Because c-Fos is often used as a proxy for neuronal firing, this result raises questions about the generalizability of RMTg responses to other aversive stimuli, albeit with the caveat that c-Fos likely reflects intracellular signaling events rather than firing per se (*Kovács, 2008*).

In addition to their limited range of tested stimuli, earlier studies had also failed to characterize the influence of RMTg neurons on DA responses to aversive stimuli, raising further questions about the proposed RMTg role. Increasingly many studies have noted that DA responses to aversive

stimuli, although somewhat complex, often exhibit a prominent inhibitory component (*Fiorillo et al., 2013*; *Matsumoto et al., 2016*; *Tian and Uchida, 2015*), and that the aversive stimulus-induced inhibition is associated with behavioral avoidance and diminished learning (*Chang et al., 2016*; *Chang et al., 2018*; *Lammel et al., 2012*). Previous study has suggested that these inhibitory responses are primarily driven by GABAergic transmissions onto DA neurons (*Henny et al., 2012*). However, the sources of the GABAergic transmissions are still unknown, and have been proposed to arise from numerous possible sources including not only the RMTg, but also the lateral hypothalamus, ventral pallidum, extended amygdala, and GABAergic neurons in the VTA (*Jennings et al., 2013*; *Tian et al., 2016*). Hence, the specific RMTg contribution to VTA responses is not known.

To address these questions about generalizability of RMTg responses to aversive stimuli and its influence on DA neurons, we recorded VTA and RMTg neuron responses to a wide range of aversive stimuli in freely-moving rats, while using excitotoxic lesions to investigate the RMTg influence on VTA neuron firings and on conditioned place aversion to these stimuli.

## Results

### RMTg neurons are activated by diverse phasic aversive stimuli

To examine the generalizability of RMTg responses to aversive stimuli, we recorded RMTg neuron responses to six distinct aversive stimuli covering a broad range of stimulus modalities in animals that had been previously trained to distinguish auditory cues predicting reward delivery or nothing (*Figure 1A,B*). Three of these stimuli were phasic: footshock (0.7 mA), loud siren (115 dB), and bright light (1600 lumens), which lasted for 10 ms, 1 s, and 2 s respectively. Three additional stimuli were sustained: lithium chloride (LiCl), restraint stress, and cocaine, lasting for several minutes in duration (*Figure 2A*). Stimuli were chosen to represent distinct sensory modalities, whose aversive qualities had been noted previously (*Ettenberg et al., 1999*; *Tzschentke, 2007*; *Winston et al., 2001*). All recorded neurons were also tested for their response to reward cues and neutral tones (75 dB) that were followed by a sucrose pellet delivery and no consequence, respectively.

Out of 151 recorded neurons, 59 were located in the RMTg, as defined by immunostaining for FOXP1 (*Lahti et al., 2016*; *Smith et al., 2018*) (*Figure 2—figure supplement 1A,B*). Consistent with previous studies that RMTg neurons encode motivational valence (*Hong et al., 2011*; *Jhou et al., 2009*), we found that RMTg neurons on average showed significant inhibition to reward cues during a time window 200–400 ms post-stimulus, and rapid excitations to all other phasic stimuli 0–100 ms post-stimulus (p < 0.0001 for all stimuli) (*Figure 2B*; *Figure 2C*). Moreover, responses to the three phasic aversive stimuli (footshock, siren, and bright light) were significantly greater than responses

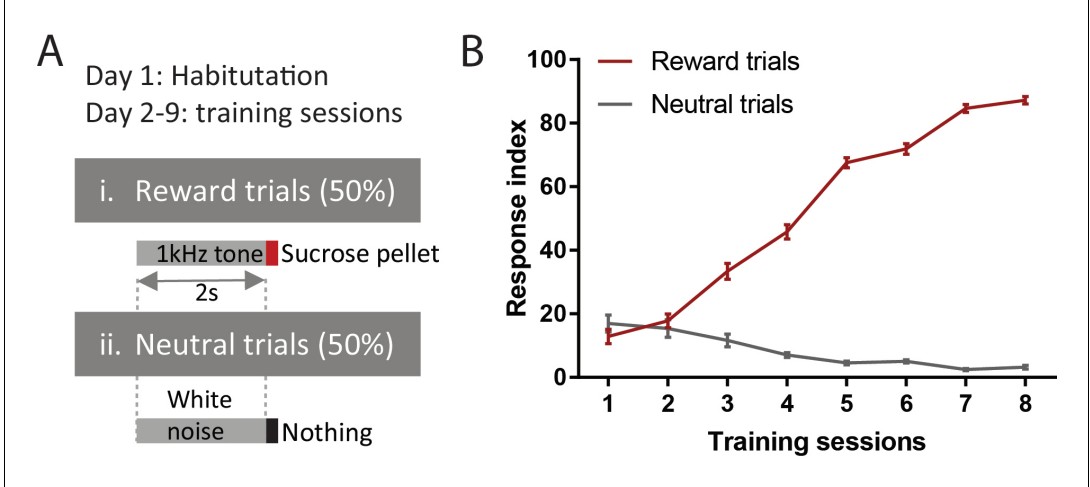

**Figure 1.** Behavioral training procedure. (**A**) Schematic of training paradigm. (**B**) Training performance. Response index is the percentage of trials in which animals made nose pokes within 2 s of onset of reward or neutral cues.
DOI: https://doi.org/10.7554/eLife.41542.002

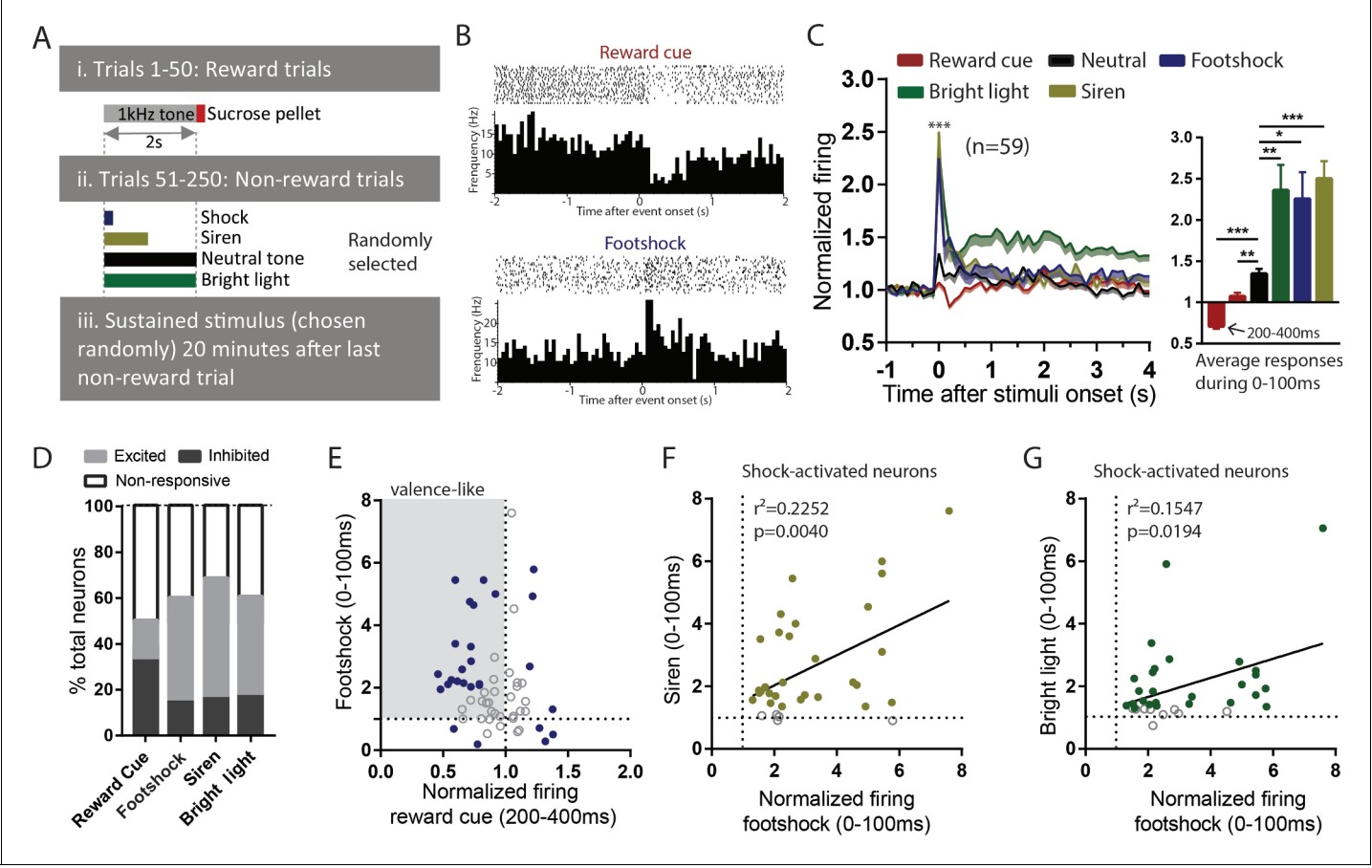

**Figure 2.** RMTg neurons are activated by diverse phasic aversive stimuli. (**A**) Schematic of recording paradigm. (**B**) Raster plots of a representative RMTg neuron response to reward cues and footshocks. (**C**) RMTg neurons on average showed inhibition to reward-predictive cues during a 200–400 ms post-stimulus window, and small excitations to neutral tones and large excitations to footshock, siren, and bright light during a 0–100 ms post-stimulus windows. (**D**) Percentage of RMTg neurons that showed inhibition, excitation or no response to stimuli (200–400 ms window for reward cues, and 0–100 ms window for aversive or neutral stimuli). (**E**) Scatterplot of individual neurons' responses to reward cues and footshocks. Many reward-cue inhibited neurons were also excited by footshocks, consistent with a valence-encoding pattern. Blue solid dots: neurons significantly responding to both reward cues and footshocks. Gray shaded box: neurons inhibited by reward cues and excited by footshocks, consistent with hypothesized valence-encoding. (**F, G**) RMTg neurons activated by footshocks tended to also be activated by siren and bright light, in proportion to the magnitude of response to footshock. Solid dots: neurons significantly responding to siren and bright light. * indicates p < 0.05, ** p < 0.01, *** < 0.0001.

DOI: https://doi.org/10.7554/eLife.41542.003

The following figure supplement is available for figure 2:

**Figure supplement 1.** Histology of RMTg recordings.
DOI: https://doi.org/10.7554/eLife.41542.004

to the neutral tone, suggesting an aversion-related augmentation in RMTg responses to aversive stimuli (F = 1.582, p = 0.027, p = 0.0001, and p = 0.0048 for neutral tone compared with footshock, siren, and bright light, repeated measures one-way ANOVA, Holm-Sidak test for multiple comparison) (*Figure 2C*). Notably, even though the duration of the aversive stimuli ranged from 10 ms up to 2 s, RMTg neuron response durations were largely confined to a period 0–100 ms after stimulus onset, independently of the total stimulus duration.

When individual neurons (instead of population averages) were analyzed, we observed considerable variation among individual neuron responses to these stimuli. Specifically, we found 51% (30/59) of RMTg neurons showing significant changes relative to baseline firing (in either direction) to reward cue during a window 200–400 ms post-stimulus, with almost three-fourths of responsive neurons (22/30 = 73%) showing inhibition, and the remaining showing excitation. Furthermore, 59% (35/59), 70% (41/59), and 61% (36/59) of RMTg neurons showed significant changes from baseline firing

(in either direction) to footshock, siren, and bright light during a window 0–100 ms post-stimulus, again with roughly three-fourths showing excitations (24/35 = 68%, 31/41 = 75%, and 26/36 = 72% for footshock, siren, and light, respectively) (*Figure 2D*). Interestingly, we noticed that a majority (14/22) of reward cue-inhibited neurons showed significant excitations to footshock, even though shock-excited neurons are only 41% of all RMTg neurons, indicating that shock-excitation patterns are particularly concentrated among the reward-cue inhibited population, contributing to an overall valence-encoding pattern in this population (*Figure 2E*). Moreover, among RMTg neurons significantly excited by footshock, these responses were positively correlated with responses of each individual neuron to siren and bright light, indicating that neurons more strongly activated by the footshock were also more strongly activated by siren and bright light ($r^2 = 0.2252$, $p = 0.004$ and $r^2 = 0.1547$, $p = 0.0194$ for siren and light respectively, 0–100 ms post-stimulus) (*Figure 2F,G*).

## RMTg neurons exhibit biphasic responses to sustained aversive stimuli consistent with opponent process theory

In addition to testing RMTg responses to multiple phasic aversive stimuli, we also examined these same neurons' responses to one of several sustained aversive stimuli lasting several minutes each. These were: a low dose of LiCl (10 mg/kg i.p.), restraint stress (6 min), or cocaine (0.75 mg/kg i.v.), which previous studies had shown produce rewarding effects for about 10 min followed by an aversive 'crash' beginning around 15 min post-injection (*Ettenberg et al., 1999*; *Jhou et al., 2013*). All stimuli were administered roughly 20 min after rats had been tested with phasic stimuli (*Figure 2A*), allowing subsequent comparison of neural responses across multiple stimuli. Notably, the dose of LiCl that we used is relatively low compared to the much higher doses commonly used to induce conditioned taste aversions, and is thought to produce modest aversive effects lasting only approximately 15 min (*Tomasiewicz et al., 2006*).

All these sustained stimuli produced two distinct phases of response in RMTg firing. We found that both LiCl (n = 22) and restraint stress (n = 12) increased RMTg firing for several minutes after the onset of each stimulus (first 0–10 min and 0–3 min post-stimulus, respectively, $p = 0.01$ and $p = 0.02$, repeated measure one-way ANOVA, Holm-Sidak test for multiple comparison). These periods of activation are somewhat shorter than the expected duration of the aversion for each stimulus (15 and 6 min, respectively), suggesting some habituation of the RMTg excitation even while the stimulus is still ongoing. Interestingly, RMTg firing exhibiting a 'rebound' inhibition to below baseline levels just after the offset of the aversive phases of each stimulus (20–30 min post-stimulus for LiCl and 9–12 min for restraint stress, $p = 0.047$, and $p = 0.036$, respectively, repeated measure one-way ANOVA, Holm-Sidak test for multiple comparison) (*Figure 3A,E*). In contrast, saline injections of equal volume as the LiCl injections had no effect on RMTg firing during either of the two time windows where LiCl had produced responses (n = 12, $p = 0.23$ and $p = 0.84$, respectively) (*Figure 3A*). As with phasic stimuli, analysis of individual neurons showed heterogeneous responses to LiCl and restraint stress. Roughly 36–42% of RMTg neurons showed both an initial excitation and a rebound inhibition (8/22 and 5/12 neurons for LiCl and restraint stress, respectively), while the remaining neurons typically showed responses during only one of the two phases (*Figure 3B,F*). We conducted further analyses to examine whether responses during initial and rebound phases were consistent with encoding of motivational states. We found that RMTg responses during the initial phase of LiCl and restraint stress were positively correlated with responses of these same neurons to footshock ($r^2 = 0.1469$, $p = 0.0033$ and $r^2 = 0.3783$, $p = 0.033$, respectively) (*Figure 3C,G*), while their rebound inhibitory phase responses correlated with responses to reward cues ($r^2 = 0.2165$, $p < 0.0001$ and $r^2 = 0.3954$, $p = 0.038$, respectively) (*Figure 3D,H*).

As LiCl and restraint stress are both aversive, we hypothesized that the rebound inhibitions of RMTg neurons may be correlated with the rewarding 'relief' upon the removal of these stimuli. Conversely, we hypothesized that the converse might also be true, that RMTg neurons might show a rebound *excitation* after removal of a sustained rewarding stimulus, as we had observed earlier in the LHb (*Jhou et al., 2013*). Thus, we further examined RMTg responses to a 0.75 mg/kg i.v. cocaine infusion. We found that RMTg neurons (n = 38) showed similar bi-phasic responses to cocaine as to LiCl and restraint stress, but in the opposite direction. Specifically, they were initially inhibited by cocaine 0–10 min post-infusion (initial phase), and subsequently activated 20–30 min post-infusion (rebound phase), when cocaine was aversive ($p = 0.0001$ and $p = 0.038$, repeated measure one-way ANOVA, Holm-Sidak test for multiple comparison). Saline infusions (n = 14) had no effect on RMTg

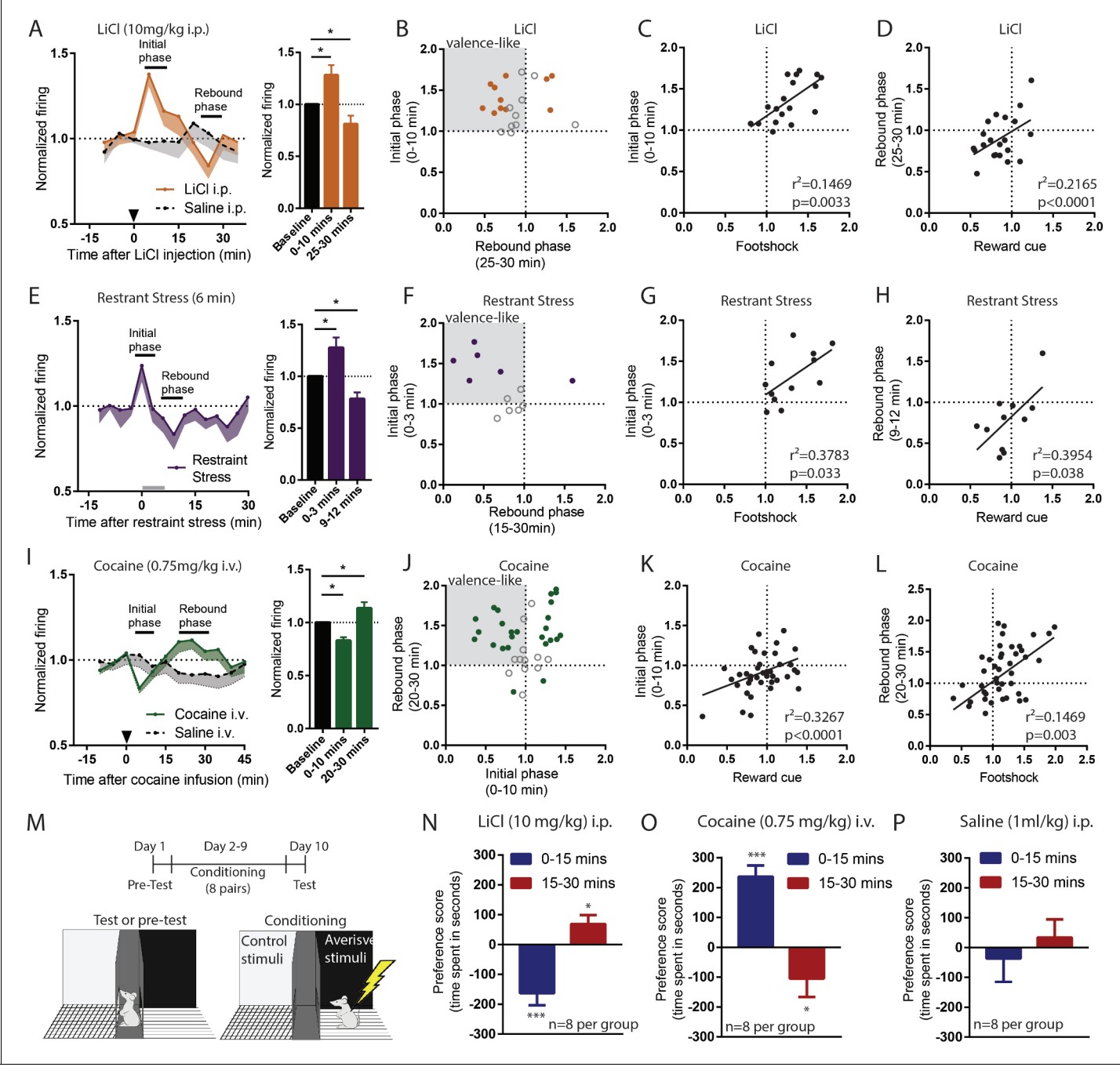

**Figure 3.** RMTg neurons exhibit biphasic responses to sustained aversive stimuli consistent with opponent process theory. (**A**, **B**) RMTg neurons showed activation to a low dose of LiCl for 10 min post injection, and (**E**, **F**) to restraint stress during the first 3 min of the 6 min restraint. Both aversive stimuli also produced a rebound inhibition of firing below baseline during a later time window (20–30 min window for LiCl, and 9–12 min window for restraint stress). (**C**, **D**) Individual neuron responses during initial phases of LiCl and restraint stress correlated with their responses to footshock, while responses during rebound phases correlated with responses to reward cue (**G**, **H**). (**I**, **J**) Cocaine infusion produced an opposing pattern in RMTg neurons, with inhibition during the first 10 min followed by a rebound excitation 15–25 min post infusion. (**K**, **L**) Individual responses during rebound (aversive) phase of cocaine were correlated with their responses to footshock, while responses during initial (rewarding) phase were correlated with their responses to reward cue. (**M**) Schematic of conditioned place preference regimen. (**N**) Low dose of LiCl (10 mg/kg) i.p. injection induced place aversion during 0–15 min and place preference during 15–30 min post-stimulus. (**O**) Cocaine (0.75 mg/kg) i.v. infusion induced place preference during 0–15 min and place aversion 15–30 min post-stimulus. (**P**) Saline injection did not produce place preference during either time window. Solid dots in **B**, **F**, and **J**: neurons significantly responding during both initial and rebound phases.

DOI: https://doi.org/10.7554/eLife.41542.005

firing during either phase (p = 0.62 and p = 0.25) (*Figure 3I*). Furthermore, when individual neuron responses were analyzed, we found that 32% of RMTg neurons (12/38) exhibited both inhibition during the initial phase and excitation during the rebound phase, again consistent with the negative valence-encoding pattern seen previously (*Figure 3J*). Furthermore, responses during the initial phase were positively correlated with responses of these same neurons to food-predictive cues ($r^2$ = 0.3267, p < 0.0001), while responses during the rebound phase were positively correlated with responses to footshocks ($r^2$ = 0.1469, p = 0.003).

Although these results imply that RMTg neurons bi-directionally encode aversive and rewarding properties of sustained stimuli, they do not indicate whether the early and rebound phases of LiCl and cocaine are indeed aversive and rewarding. Thus, we tested four groups of animals (eight per group) for conditioned place preference or aversion to the same doses of LiCl and cocaine as during the recordings. We placed animals into conditioning chambers either 0–15 min or 15–30 min post-stimulus, to align with the two phases of neural responses seen in earlier recordings (*Figure 3M*). In separate groups of LiCl-treated groups, we found that animals developed place aversion when placed into conditioning chambers immediately after the injection, but showed place preference when placed into chambers 15 min after the injection (p < 0.0001 and p = 0.02 for 0–15 min and 15–30 min, respectively) (*Figure 3N*). In contrast, separate groups of cocaine-treated animals developed place preference if conditioned immediately after infusion, and place aversion if conditioned 15 min later (p = 0.034 and p < 0.0001, respectively) (*Figure 3O*). Saline injections did not produce place preference during either time window (p = 0.412 and p = 0.562, respectively) (*Figure 3P*). Together, our results indicate that bidirectional behavioral effects of sustained rewarding or aversive stimuli correlate with bidirectional responses of RMTg neurons to these stimuli, suggesting a possible substrate for an 'opponent' responses to stimuli described decades ago by opponent process theory (*Solomon and Corbit, 1973*).

## VTA-projecting RMTg neurons preferentially show valence-encoding patterns

As previous results have indicated heterogeneities in RMTg responses to both phasic and sustained stimuli, with only 30–40% of RMTg neurons encoding negative valence, we next examined whether the heterogeneity in response patterns might be related to heterogeneity in RMTg projection targets. Thus, we used endoscopic calcium imaging and selectively recorded RMTg neurons that project to either the VTA or the dorsal raphe nucleus (DRN), both of which are implicated in encoding of motivational stimuli (*Cohen et al., 2012*; *Li et al., 2016*). Because of the need to keep GRIN lenses short, this experiment was performed in mice, rather than rats, but the anatomy of the RMTg is very similar between species (*Smith et al., 2018*). We injected into wild type mice a retrogradely transported canine adenovirus expressing Cre recombinase (CAV2-Cre) into either the VTA or (in separate mice) the DRN, along with a second virus into the RMTg expressing a Cre-dependent fluorescent calcium indicator (gCaMP6f) (*Figure 4A–C*, *Figure 4—figure supplement 1A,B*). Mice were tested in separate sessions in which they received either a footshock (0.3mA), or an auditory tone followed by food pellet delivery.

We found that VTA-projecting neurons were on average inhibited by reward cues and activated by footshocks, that is a negative valence-encoding pattern (*Figure 4D*, *Figure 4—figure supplement 1C,D*). Analysis of individual neurons further confirmed this; among 25 VTA-projecting neurons, a majority (14/25 neurons) were inhibited by reward cues and activated by footshocks, while 7/25 neurons responded to only one of the two stimuli, with the remaining four neurons showing no response to either stimulus. Furthermore, individual neuron responses to reward cues correlated negatively with responses to footshocks ($r^2$ = 0.2661, p = 0.0070) (*Figure 4F*).

In marked contrast to VTA-projecting neurons, DRN-projecting neurons on average showed no responses to reward cues, although there was an average activation by footshocks, similar to VTA-projecting neurons (*Figure 4E*). Analysis of individual neurons further showed that responses to reward cues and footshocks did not correlate with each other (p = 0.9775) (*Figure 4G*). Overall, VTA- and DRN-projecting neurons exhibited markedly different proportions of neurons that were inhibited by reward cues (p = 0.001, Chi-square) (*Figure 4H*). Thus, our results indicate that VTA-projecting but not DRN-projecting RMTg neurons appear highly enriched in negative valence-encoding patterns.

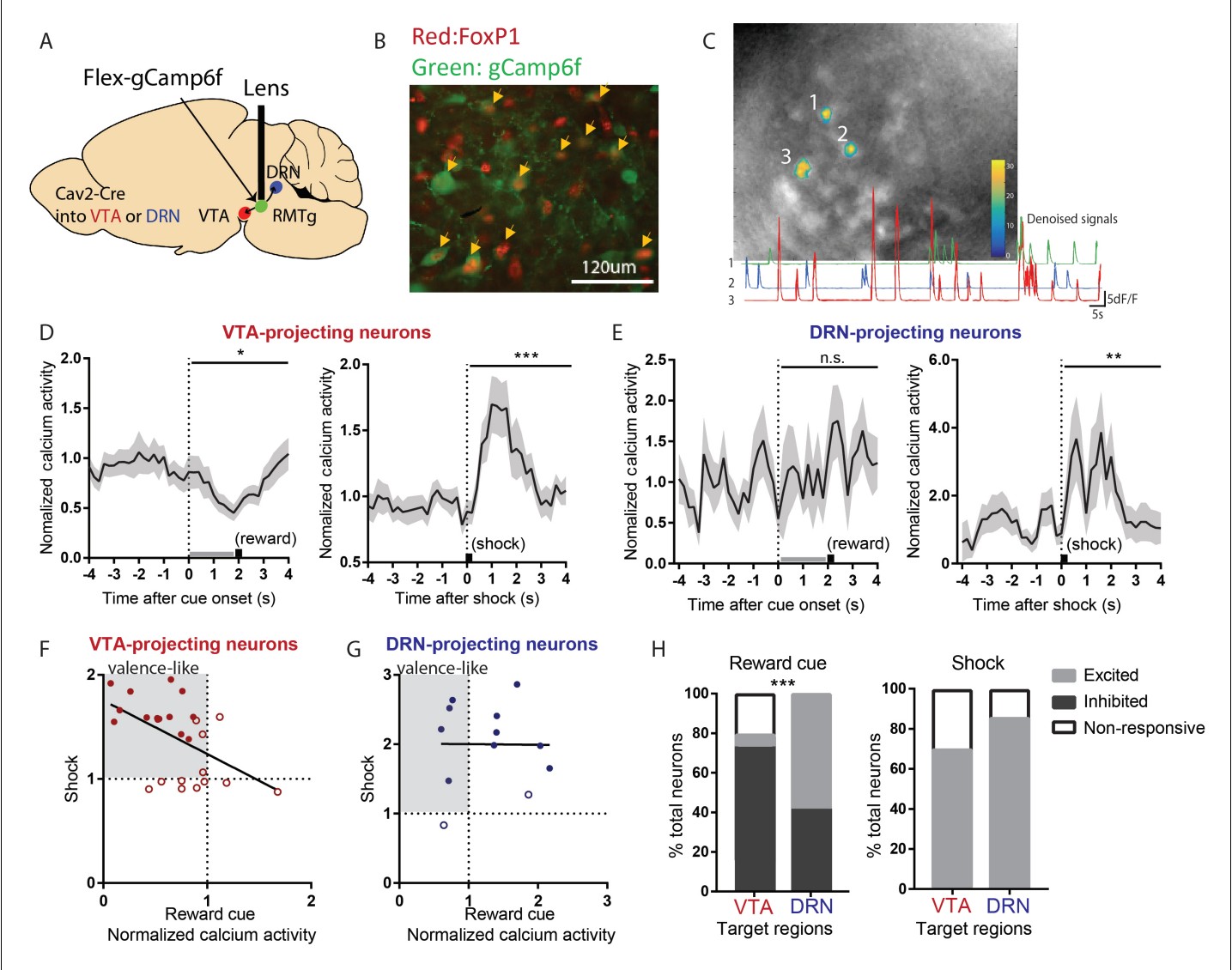

**Figure 4.** VTA-projecting RMTg neurons preferentially show valence-encoding patterns. (**A**) Cav2-cre injected into the VTA or DRN was retrogradely transported to the RMTg in mice, driving gCaMP6f expression in subsets of RMTg neurons projecting to the VTA or DRN respectively. (**B**) Representative photograph of the RMTg region in which gCaMP6f in RMTg (green label) is co-expressed with FOXP1 (red), a transcription factor locally specific to RMTg neurons. (**C**) A representative photo of gCaMP6f positive neurons in vivo (upper panel) and denoised Ca²⁺ traces extracted from the marked neurons (lower panel). (**D**) VTA-projecting RMTg neurons showed an average inhibition by reward cues, and excitation by footshocks. (**E**) DRN-projecting neurons showed no average response to reward cues, but were excited by footshocks. (**F**) Among VTA-projecting RMTg neurons, neurons showing stronger excitations to shock tended to also show stronger inhibitions to the reward cue, while individual DRN neurons did not show this correlation (**G**). Colored solid dots: neurons significantly responding to both reward cues and footshocks. (**H**) VTA-projecting neurons were much more likely to be inhibited by the reward cue than DRN-projecting neurons, and much less likely to be activated, while VTA- and DRN-projecting neurons were both predominantly activated by shock.

DOI: https://doi.org/10.7554/eLife.41542.006

The following figure supplement is available for figure 4:

**Figure supplement 1.** Histology of endoscopic calcium imaging and individual responses of VTA-projecting RMTg neurons.

DOI: https://doi.org/10.7554/eLife.41542.007

## Excitotoxic RMTg lesions abolished VTA inhibitions by aversive stimuli

Our findings that RMTg neurons are activated by aversive stimuli and VTA-projecting neurons are preferentially negative valence-encoding suggest that the RMTg could drive VTA responses to aversive stimuli, a hypothesis we tested by recording VTA neuron responses to phasic aversive stimuli

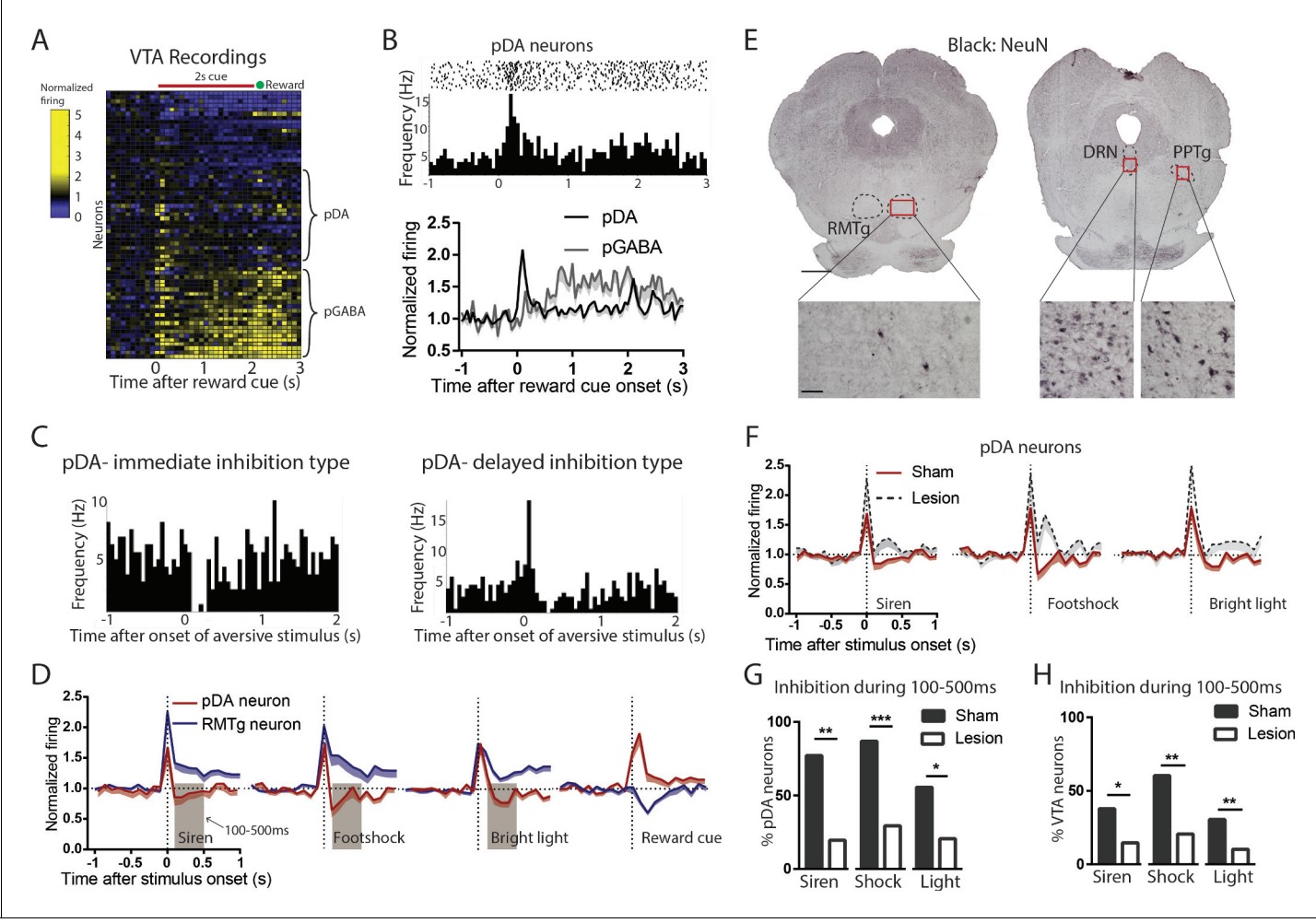

**Figure 5.** Excitotoxic RMTg lesions abolished VTA inhibitions by aversive stimuli. (**A, B**) Heatmap showing all VTA neuron responses to reward cues in sham group. pDA neurons in the VTA were classified by their phasic activation to reward cues (0–200 ms post-stimulus window), while pGABA neurons were classified by the presence of sustained activations (200–2000 ms post-stimulus window). (**C**) Raster plots of immediate inhibition and delayed inhibition response types observed in pDA neuron after aversive stimuli. (**D**) Comparisons of RMTg (blue trace) and pDA (red trace) neuron responses to affective stimuli. All three aversive stimuli elicited initial excitations in both RMTg and pDA neurons, after which RMTg neurons remained excited while pDA neurons showed inhibition during 100–500 ms window post-stimulus (brown-shaded boxes). pDA neurons were activated by reward cues, and at faster latencies than RMTg inhibition to the same cue, making it unlikely that pDA activations to the reward cue would be driven by the RMTg. (**E**) NeuN staining showed that lesions were mostly restricted to the RMTg area and did not extend to surrounding structures such as the pedunculopontine nucleus (PPTg) and dorsal raphe nucleus (DRN). Scalebars: 1 mm and 100 μm for left and right panels. (**F**) RMTg lesion (dashed trace) eliminated aversion-induced inhibition in pDA neurons. (**G**) Bar graphs again showing loss of aversion-induced inhibition during 100–500 ms in pDA neurons after RMTg lesions. (**H**) Loss of aversion-induced inhibition in all recorded VTA neurons.

DOI: https://doi.org/10.7554/eLife.41542.008

The following figure supplements are available for figure 5:

**Figure supplement 1.** Histology of VTA recordings.

DOI: https://doi.org/10.7554/eLife.41542.009

**Figure supplement 2.** Supplementary VTA recordings.

DOI: https://doi.org/10.7554/eLife.41542.010

with or without RMTg lesions (*Figure 5E*, *Figure 5—figure supplement 1*). NeuN staining showed that lesions were mostly restricted to the RMTg area and did not extend to surrounding structures such as the pedunculopontine (PPTg) and dorsal raphe nuclei (DRN) (*Figure 5E*). We classified recorded VTA neurons as being putative dopamine neurons (pDA neurons) by their phasic

activations to reward cues (Sham: n = 21 pDA, n = 45 non-pDA, Lesion: n = 17 pDA, n = 40 non-pDA, p < 0.05 during 0–200 ms after reward cues), an activity signature repeatedly shown to correlate highly with optogenetically identified DA neurons in mice (*Cohen et al., 2012*; *Eshel et al., 2015*; *Matsumoto et al., 2016*) (*Figure 5A,B*). We also observed about 1/3 of recorded VTA neurons in sham lesioned rats exhibiting ramping activities in responses to reward cues (p < 0.05 during 200–2000 ms after reward cue onset), which are consistent with firing patterns found in genetically identified GABAergic neurons in the VTA (*Cohen et al., 2012*; *Eshel et al., 2015*) (*Figure 5A,B*). Additionally, 8 of the 66 recorded VTA neurons in sham group showed inhibitory responses to reward cues.

We found that most pDA neurons in the sham group showed inhibitions to aversive stimuli, in some cases after a brief initial excitation (*Figure 5C*). Specifically, 86% of stimulus-responsive pDA neurons were inhibited by footshock (54% showing only inhibition; 32% showing delayed inhibition after a brief initial excitation), 75% by siren (28% showing only inhibition; 47% showing delayed inhibition), and 56% by bright light (27% showing only inhibition; 29% showing delayed inhibition), while the remaining showed excitation only (*Figure 5—figure supplement 2A–C*). Notably, inhibition by aversive stimuli was most prominent during a 100–500 ms window post-stimulus, while activation by reward cues was most prominent in an earlier 0–200 ms window post-stimulus, a timing pattern opposite to that of RMTg neurons in which responses to aversive stimuli tended to be faster by a few hundred milliseconds (*Figure 5D*).

Among recorded VTA neurons, we found that RMTg lesions did not affect the magnitude nor the percentage of neurons responding to reward cues (0–200 ms post-stimulus) (p > 0.05, two-way ANOVA, Bonferroni test for multiple comparison, p = 0.879, Chi-square) (*Figure 5—figure supplement 2D,E*). Hence, we further analyzed reward-activated VTA neurons on the presumption that these were pDA in both lesioned and unlesioned animals. We found that, after RMTg lesions, pDA neurons no longer showed an average inhibition to any of the three phasic aversive stimuli (*Figure 5F*), and the proportions of pDA neurons inhibited by aversive stimuli were dramatically reduced to 19%, 16%, and 18% for footshock, siren, and bright light, respectively (p < 0.0001, p = 0.008, and p = 0.048, Chi-square, 100–500 ms post-stimulus) (*Figure 5G*). Due to the potential inaccuracy of our classification of pDA neurons, we then analyzed all recorded VTA neurons instead of pDA neurons alone. Consistent with pDA analysis, the proportions of inhibitory responses were also reduced by RMTg lesions, although to a less extreme degree than the pDA population. Specifically, in intact animals, 59%, 42%, and 36% of VTA neurons showed significant inhibitions to footshock, siren, and bright light, while in RMTg-lesioned rats, these proportions were reduced to 21%, 16%, and 9%, respectively (p = 0.005, p = 0.017, and p = 0.004, for footshock, siren, and bright light, respectively, Chi-square) (*Figure 5H*).

While pDA inhibitory responses to aversive stimuli were reduced by RMTg lesions, the proportions of pDA neurons showing pure excitation were either marginally increased or unaffected (p = 0.182, p = 0.047, and p = 0.766 for siren, footshock, and bright light, respectively, Chi-square) (*Figure 5—figure supplement 2G*). As noted above, many pDA neurons that were inhibited by aversive stimuli did so after brief initial excitations, and RMTg lesions increased the magnitudes of these initial excitations relative to shams (p = 0.013, p = 0.021, and p = 0.009 for siren, shock, and bright light, respectively, unpaired t-test), without affecting the proportion of pDA neurons exhibiting them (p = 0.169, p = 0.328, and p = 0.126 for siren, shock, and bright light, respectively, Chi-square) (*Figure 5—figure supplement 2H*). Notably, RMTg lesions did not alter basal firing rates of pDA neurons (5.4 Hz vs 5.2 Hz, n = 23, p = 0.507, unpaired t-test). However, we did find that RMTg lesions increased the percentage of spikes found in bursts of pDA neurons (p = 0.014, unpaired t-test) (*Figure 5—figure supplement 2F*). Together, RMTg lesions influence both inhibitions and excitations of pDA neurons to aversive stimuli, greatly reducing inhibitions, and modestly increasing excitations to these stimuli, while having no apparent effect on responses to reward predictive cues nor neutral tones.

## RMTg lesions disrupt conditioned place aversion to a wide range of stimuli

Finally, we examined the effect of RMTg lesions on conditioned place preference for three of the aversive stimuli that we tested earlier: siren, bright light, LiCl, and the delayed aversive phase of cocaine (*Figure 6A*, *Figure 6—figure supplement 6–*). Using a three-chambered apparatus, we

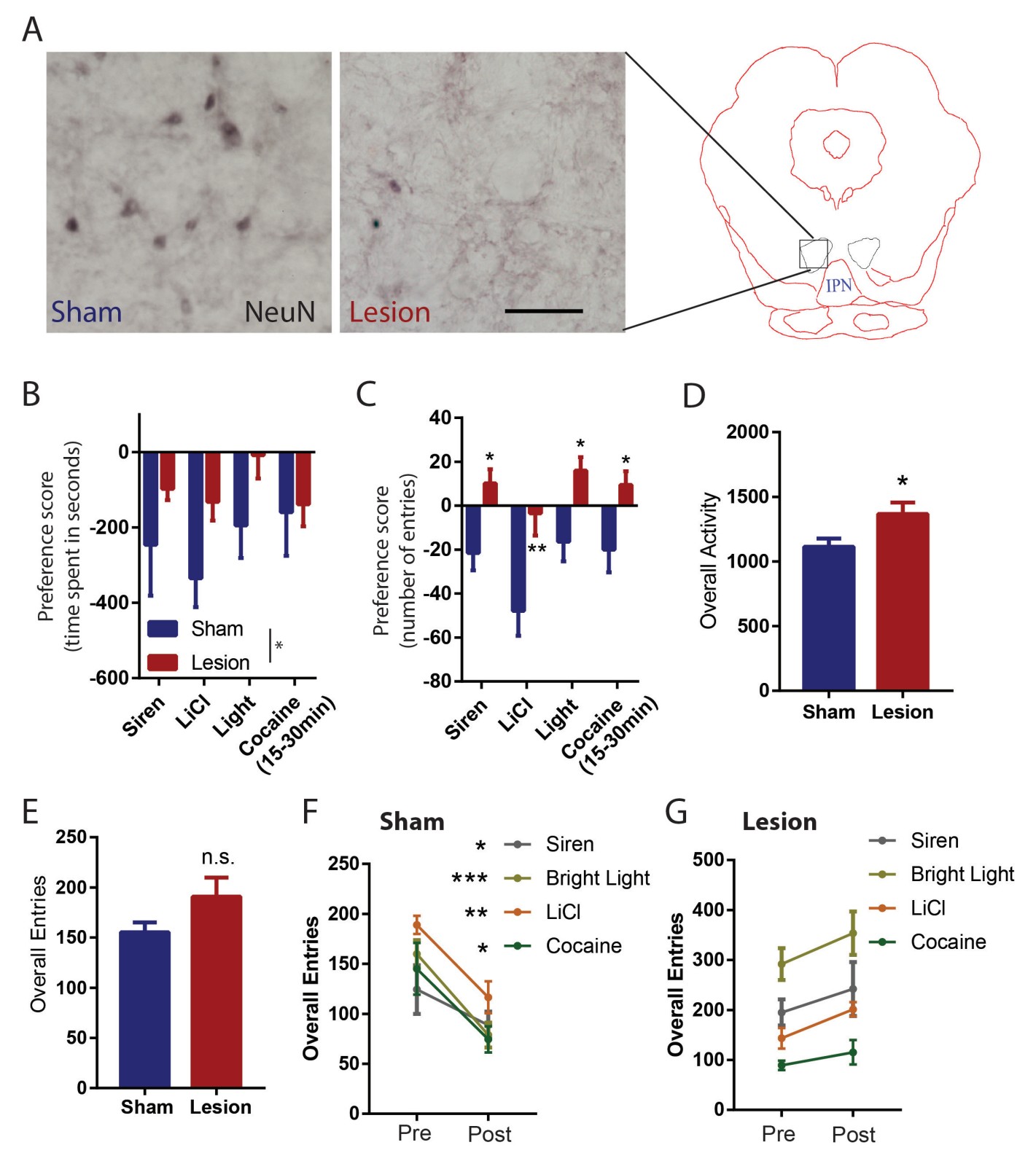

**Figure 6.** RMTg lesions disrupt conditioned place aversion to a wide range of stimuli. (**A**) Photographs of immunostaining of NeuN in RMTg region with and without excitotoxic lesions. Scalebar: 100 μm. RMTg-lesions reduced place aversion scores as measured by relative time spent in stimulus-paired versus unpaired chambers (**B**), and abolished place aversion as measured by relative entries into paired versus unpaired chambers (**C**). (**D**) RMTg lesions did not alter total entries into chambers. (**E**) RMTg-lesioned rats lacked the training-induced reduction in locomotion seen in shams (**F, G**).

*Figure 6 continued on next page*

*Figure 6 continued*

DOI: https://doi.org/10.7554/eLife.41542.011

The following figure supplement is available for figure 6:

**Figure supplement 1.** Excitotoxic lesion placements in the RMTg.

DOI: https://doi.org/10.7554/eLife.41542.012

were able to measure the effect of conditioning on both the time spent in each chamber (stimulus-paired, unpaired, and neutral) as well as the relative number of entries into the paired and unpaired chambers (*Figure 3M*). We found that on average, sham rats expressed a significant aversion to all stimuli, as measured by both time spent in, and entries into, the stimulus-paired versus unpaired chambers (time spent: $p = 0.007$, $p < 0.0001$, $p = 0.039$, and $p = 0.05$, number of entries: $p = 0.014$, $p < 0.0001$, $p < 0.0001$, and $p = 0.05$, respectively for siren, LiCl, bright light, and cocaine, z-score).

A two-way ANOVA comparing lesioned versus sham groups then showed a significant main effect of lesion for time spent in paired chamber ($p = 0.01$) (*Figure 6B*), due to lesioned rats having a smaller bias away from the stimulus-paired chambers, compared to sham rats. When examining specific stimuli, the lesion-induced reduction was not significant for any particular stimulus ($p = 0.216$, $0.216$, and $p = 0.216$ for siren, bright light and LiCl, respectively, $p = 0.87$ for cocaine, Bonferroni test for multiple comparisons) (*Figure 6B*). RMTg lesions produced a somewhat stronger effect on the preference score as measured by entries, with the two-way ANOVA again showing a main effect of lesion ($p < 0.0001$), with this effect also being significant for each stimulus ($p = 0.027$, $p = 0.003$, $p = 0.021$, and $p = 0.043$ for siren, LiCl, bright light, and cocaine, respectively) (*Figure 6C*). Although preference scores could be confounded if RMTg lesions produce extreme motoric disinhibition, we found this was not the case; in particular, while the lesion group showed slightly greater overall locomotor activities measured by photobeam counts ($p = 0.02$, unpaired t-test) (*Figure 6D*), lesion and sham groups did not show differences in total chamber entries prior to conditioning, suggesting that effects of RMTg lesions on locomotion were minor ($p = 0.106$, unpaired t-test) (*Figure 6E*). Interestingly, aversive conditioning reduced total chamber entries for all three stimuli in the sham group ($p = 0.04$, $p = 0.0004$, $p = 0.001$, $p = 0.035$ for siren, bright light, LiCl, and cocaine, respectively, paired t-test) (*Figure 6F*), but not the RMTg lesioned group ($p = 0.34$, $p = 0.22$, $p = 0.01$, $p = 0.425$ for siren, bright light LiCl, and cocaine, respectively, paired t-test) (*Figure 6G*), suggesting an overall inhibitory effect of aversive conditioning in normal rats that was not present in the lesioned group.

## Discussion

We showed that in addition to being inhibited by reward-predictive cues, that RMTg neurons are activated by six distinct aversive stimuli of widely varying sensory modalities and timescales, while also playing key roles in driving VTA inhibitions and behavioral avoidance to aversive stimuli. Furthermore, we found evidence that removal of sustained aversive stimuli (LiCl and restraint stress) produces reward-correlated responses in the RMTg, while removal of a sustained reward (cocaine) produces aversion-correlated responses in the RMTg. These RMTg responses parallel actual reward and aversion as measured by conditioned place preference, and suggest possible neural substrates of the opponent process model proposed decades ago to describe the phenomenon that removal of a strong affective stimulus often produces an affective state in the opposite direction of the initial stimulus (*Solomon and Corbit, 1973*).

### Broad encoding of valence in RMTg contributes to VTA valence encoding

The current study established a broad role for the RMTg in driving VTA inhibitions to aversive stimuli. Except for footshocks and shock cues (*Jhou et al., 2009*), RMTg responses to other aversive stimulus modalities had been largely untested, with the exception of one study showing that many aversive stimuli do *not* induce c-Fos activation in the RMTg (*Sánchez-Catalán et al., 2017*). Additionally, previous studies had also identified several brain regions other than the RMTg that might also contribute to DA responses to aversive stimuli including ventral pallidum and hypothalamus

(*Nieh et al., 2016*; *Tian et al., 2016*; *Tian and Uchida, 2015*). Given the uncertainty of RMTg generalizability to a wide range of aversive stimuli and it influence on DA responses, it is notable that we saw RMTg activations by all six aversive stimuli that we tested (including several that did not induce c-Fos in the earlier study of Sanchez-Catalan et al.), and that RMTg activation to phasic stimuli preceded pDA inhibition to the same stimuli by a few hundred milliseconds, consistent with a possible causal RMTg role in driving DA inhibition to these stimuli. Furthermore, we found that RMTg lesions greatly reduced aversion-induced inhibitions to all tested stimuli in either classified pDA neurons or the entire VTA population.

Although our recorded pDA neurons were not identified genetically (e.g. by optogenetic photo-tagging), we used a classification method involving responses to reward-predictive cues that in mice is highly correlated with optogenetically identified DA neurons (*Cohen et al., 2012*; *Matsumoto et al., 2016*; *Tian and Uchida, 2015*). Moreover, we found that the aversion-induced inhibitory patterns were enriched in pDA neurons compared to all recorded VTA neurons, consistent with previous studies showing that DAergic neurons are the predominant cell-type exhibiting inhibitory responses to aversive stimuli in the VTA (*Brischoux et al., 2009*; *Cohen et al., 2012*; *Ungless et al., 2004*). Hence, we speculate that a majority of recorded pDA neurons are DAergic. However, it is notable that a small population of vGluT2-expressing VTA neurons are found to also be excited by reward-predictive cues (*Root et al., 2018*); thus, it is possible that a small percentage of these neurons were misclassified into pDA population. However, because we noted a loss of aversive stimulus-induced inhibition across all VTA neurons, *whether or not they had been classified as pDA*, our findings that VTA encoding of aversive stimuli is RMTg-dependent does not rely on the ability to classify VTA neurons into pDA or non-pDA neurons.

## Heterogeneity and opponency of RMTg firing patterns

In RMTg neurons, negative valence-encoding patterns, that is activation by aversive stimuli and inhibition by rewarding stimuli, were the most common, but we also found other response patterns in RMTg neurons. However, valence-encoding patterns were particularly enriched in VTA-projecting but not DRN-projecting neurons, as shown by endoscopic calcium imaging in mice. Thus, our results suggest that despite the heterogeneity in RMTg response patterns, that its influence on the VTA encoding of aversive stimuli is likely to be somewhat more uniform, and consistent with the originally proposed RMTg role in driving VTA inhibitions to aversive stimuli (*Hong et al., 2011*). Although calcium transients measured in the current study may not be strictly comparable to electrophysiological firing (*Grewe et al., 2010*; *Jennings et al., 2015*), calcium transients did show clear differences between VTA- and DRN-projecting neurons, strongly indicating that heterogeneity in RMTg activation patterns are correlated with differences in RMTg projection targets. However in contrast to the valence encoding roles of the VTA-projecting RMTg population, the roles of the DRN-projecting neurons are less clear, although we found that a high percentage of DRN-projecting RMTg neurons are activated by footshock, raising the possibility that the RMTg may also drive DRN inhibition by aversive stimuli such as footshock (*Schweimer and Ungless, 2010*).

In addition to RMTg activation to aversive stimuli and its influence on VTA responses to these stimuli, we also showed evidence that RMTg neurons could dynamically encode opponent processing of motivational states. RMTg responses to all three sustained stimuli (LiCl, restraint stress, and cocaine) exhibit a delayed 'rebound' shift in the opposite direction to the initial responses. The RMTg opponent-firing responses were coincident with the opponent motivational states induced by the stimulus, suggesting parallels with opponent process theory that has been found in responses to many motivational stimuli across species (*Becerra et al., 2013*; *Ettenberg, 2004*; *Jhou et al., 2013*; *Koob et al., 1989*; *Navratilova and Porreca, 2014*; *Navratilova et al., 2012*; *Solomon, 1980*; *Solomon and Corbit, 1973*). Furthermore, previous studies have indicated an important DA role in opponent processing, as relief of pain induced increased DAergic signaling in the nucleus accumbens, and induces place preference which could be blocked by intra-VTA infusion of GABA agonists, while inhibition of mesolimbic dopamine system results in diminished relief learning (*Becerra et al., 2013*; *Mayer et al., 2018*; *Navratilova and Porreca, 2014*; *Navratilova et al., 2012*). Our results suggest that the opponent processing signals could be potentially driven by the RMTg through inhibiting and disinhibiting VTA DA neurons (*Bourdy et al., 2014*; *Jalabert et al., 2011*; *Lecca et al., 2011*; *Lecca et al., 2012*).

## Contrast between RMTg and VTA GABA neurons

The current study also provides evidence for distinct functions of RMTg and the closely adjacent VTA GABAergic neurons. This contrasts somewhat with prior studies showing similar functions, for example that they have similar inhibitory influences on DA neurons, and that activation of either RMTg or VTA GABAergic neurons induce behavioral avoidance (*Jhou et al., 2009*; *Lammel et al., 2012*; *Stamatakis and Stuber, 2012*; *Tan et al., 2012*). However, a series of elegant studies from Uchida's group have recorded genetically identified GABAergic neurons in the VTA and show that they exhibit ramping activities in responses to reward cues and that they remain activated until animals receive rewards (*Cohen et al., 2012*; *Eshel et al., 2015*). In contrast, we showed that RMTg neurons were inhibited by reward cues and then quickly returned to baseline firing levels, a completely different firing pattern from VTA interneurons in those studies. Although Uchida's studies are in mice, we also found that about 1/3 of our recorded VTA neurons exhibited similar ramping activities in response to reward cues, a response pattern almost never seen in the RMTg recordings. Furthermore, only 6% (4/66) of VTA neurons exhibited negative valence encoding (i.e. activation by aversive stimuli and inhibition by reward-predictive cues), the predominant response pattern seen in the RMTg. Furthermore, prior work noted that optogenetic inhibition of VTA GABA interneurons disrupts phasic DA responses to reward cues and results in a sustained DA activation throughout the entire presentation of cues (*Eshel et al., 2015*). In the current study, we showed that RMTg lesions removed most of the aversion-induced inhibition in VTA neurons but did not alter the proportion nor responses of VTA neurons to reward cues. Taken together, these results suggest that RMTg neurons convey completely different information onto DA neurons than VTA GABAergic neurons, and potentially mediate distinct aspects of motivated behaviors.

Notably, we observed a delay of several hundred milliseconds from the bulk of the RMTg activation to the bulk of the DA inhibition, a delay that may seem rather long for a monosynaptic connection, and certainly longer than expected for ionic conductance. However, the RMTg also expresses a number of peptides (nociceptin and somatostatin) that could mediate slower responses (*Jhou et al., 2012*; *Smith et al., 2018*).

## Behavioral significance of the RMTg

To link our electrophysiological findings to behavior, we tested the effects of RMTg lesions on conditioned place aversion to most of the aversive stimuli we tested. We used a three-chambered apparatus that allowed us to independently measure both entries to, and time spent in, the stimulus-paired and control chambers. Intact animals made fewer entries into the stimulus-paired chambers and spent less time in those chambers, consistent with these stimuli being aversive. After RMTg lesions the number of entries into each chamber no longer differed between stimulus and control, suggesting a complete loss of this particular measure of aversive bias. However, rats still spent less time in the stimulus-paired chamber than control chambers, but this difference was smaller than in unlesioned rats. Hence, RMTg-lesioned animals still appear able to exhibit a weakened behavioral bias away from the stimulus-paired chamber, but express this bias differently than unlesioned rats. In particular, unlesioned rats are less likely to enter the stimulus-paired chamber from the center chamber, but lesioned rats still leave the stimulus-paired chamber more readily than the unpaired chamber. This may be consistent with other studies in which we found deficits in the ability to inhibit behaviors associated with negative outcomes, suggesting that the primary deficit in RMTg-lesioned rats is one of impulse inhibition, rather than aversive processing per se (*Elmer et al., 2019*; *Laurent et al., 2017*; *Sánchez-Catalán et al., 2017*; *Vento et al., 2017*).

These electrophysiological findings, taken together with our behavioral effects on conditioned place test, show a broad RMTg role in processing aversive stimuli. The RMTg is activated by aversive stimuli of a remarkably wide range of modalities and timescales, is a predominant driver of VTA inhibition by such aversive stimuli, and also drives place aversion to these stimuli.

## Materials and methods

### Animals

All procedures were conducted under the National Institutes of Health Guide for the Care and Use of Laboratory Animals, and all protocols were approved by Medical University of South Carolina

Institutional Animal Care and Use Committee. Adult male Sprague Dawley rats weighing 250 to 450 g from Charles River Laboratories were paired housed in standard shoebox cages with food and water provided ad libitum until experiments started. Rats were single housed during all experiments. In total, 112 rats were used for these experiments. 95 rats completed conditioned place tests, 48 were for *Figure 3* and 47 were for *Figure 6*, of these, 24 rats had RMTg lesions. 15 rats underwent RMTg recordings, and eight underwent VTA recordings.

## Surgeries

All surgeries were conducted under aseptic conditions with rats that were under isoflurane (1–2% at 0.5–1.0 liter/min) anesthesia. Analgesic (ketoprofen, 5 mg/kg) was administered subcutaneously immediately after surgery. Rats were given at least 5 days to recover from surgery. For recording experiments, drivable electrode arrays were implanted above the RMTg (AP: −7.4 mm; ML: 2.1 mm; DV: −7.4 mm from dura, 10-degree angle) or the VTA (AP: −5.5 mm; ML: 2.5 mm; DV: −7.8 mm from dura, 10-degree angle). For lesion experiments, 50 nl of 400 mM quinolinic acid per side was injected with a glass pipette into the RMTg (AP: −7.6 mm; ML: 2.1 mm; DV: −7.8 mm from dura, 10-degree angle). Sham controls received a saline infusion of equal volume into the RMTg. Rats were kept anesthetized with pentobarbital intraperitoneally (55 mg/kg) for up to 3 hr' post-surgery to reduce excitotoxic effect.

For intravenous catheterization, rats were implanted with 0.037-inch diameter silicone tubing into the jugular vein or the femoral vein. Intravenous lines were exteriorized at the back of the neck and flushed every 2 days with sterile saline and 0.1 ml TCS lock solution to ensure patency.

Mice surgeries are described in the 'calcium imaging' section.

## Perfusions and tissue sectioning

Rats used for all experiments were sacrificed with an overdose of isoflurane and perfused transcardially with 10% formalin in 0.1M phosophate buffered saline (PBS), pH 7.4. Brains from electrophysiology experiments had passage of 100µA current before perfusion, allowing electrode tips to be visualized. Brains were removed from the skull, equilibrated in 20% sucrose solution until sunk, and cut into 40 µm sections on a freezing microtome. Sections were stored in phosphate buffered saline with 0.05% sodium azide.

## Immunohistochemistry for TH and FOXP1

Free-floating sections were immunostained for TH or FOXP1 by overnight incubation in mouse anti-TH (Millipore, MAB-377, 1: 10,000 dilution) or rabbit anti- FOXP1 (Abcam, ab16645, 1: 50,000 dilution) primary in PBS with 0.25% Triton-X and 0.05% sodium azide. Afterwards, tissue was washed three times in PBS and incubated in biotinylated donkey-anti-mouse or anti-rabbit secondary (1:1000 dilution, Jackson Immunoresearch, West Grove, PA) for 30 min, followed by three 30 s rinses in PBS, followed by 1 hr in avidin-biotin complex (Vector). For TH-staining, tissue was then rinsed in sodium acetate buffer (0.1M, pH 7.4), followed by incubation for 5 min in 1% diaminobenzidine (DAB). For FOXP1 staining, nickel and hydrogen peroxide (Vector) were added to reveal a blue-black reaction product.

For florescent staining of FOXP1, free-floating sections were incubated in rabbit anti- FOXP1 (Abcam, ab16645, 1: 50,000 dilution) primary in PBS with 0.25% Triton-X and 0.05% sodium azide. Afterwards, tissue was washed three times in PBS and incubated in cy3-conjugated donkey-anti-rabbit secondary (1:1000 dilution, Jackson Immunoresearch, West Grove, PA).

## Behavioral training for electrophysiological recordings

Rats were food restricted to 85% of their *ad libitum* body weight and trained to associate distinct auditory cues with either a food pellet or no outcome. Behavior was conducted in standard Med Associates chambers (St. Albans, VT). Food-predictive and neutral tones were a 1 kHz tone (75 dB) and white noise (75 dB), respectively. The food-predictive cue was presented for 2 s, and a food pellet (45 mg, BioServ) was delivered immediately after cue offset. The neutral tone was also presented for 2 s, but no food pellet was delivered. The two trial types were randomly presented with a 30 s interval between successive trials. A 'correct' response was scored if the animal either entered the food tray within 2 s after reward cues, or withheld a response for 2 s after neutral tones. Rats were

trained with 100 trials per session, one session per day, until they achieved 85% accuracy in any 20-trial block. Once 85% accuracy was established, rats underwent surgeries. After recovery from surgeries, rats were then trained with one extra session in which neutral tone trials were replaced by aversive trials consisting of a 2 s 8 kHz tone (75 dB) followed by a 10 ms 0.7mA footshock.

## Electrophysiological recordings

After final training, electrodes consisted of a bundle of sixteen 18 µm Formvar-insulated nichrome wires (A-M system) attached to a custom-machined circuit board. Electrodes were grounded through a 37-gauge wire attached to a gold-plated pin (Newark Electronics), which was implanted into the overlying cortex. Recordings were performed during once-daily sessions, and electrodes were advanced 80–160 µm at the end of each session. The recording apparatus consisted of a unity gain headstage (Neurosys LLC) whose output was fed to preamplifiers with high-pass and low-pass filter cutoffs of 300 Hz and 6 kHz, respectively. Analog signals were converted to 18-bit values at a frequency of 15.625 kHz using a PCI card (National Instruments) controlled by customized acquisition software (Neurosys LLC). Spikes were initially detected via thresholding to remove signals less than twofold above background noise levels, and signals were further processed using principal component analysis performed by NeuroSorter software. Spikes were accepted only if they had a refractory period, determined by <0.2% of spikes occurring within 1 ms of a previous spike, as well as by the presence of a large central notch in the auto-correlogram. Neurons that had significant drifts in firing rates were excluded. Since the shock duration used in the current study was 10 ms, the first 10 ms of data after footshock were removed in order to reduce shock artifacts.

For phasic aversive stimuli paradigm, rats were again placed on mild food deprivation, and recordings obtained in sessions consisting of 50 reward trials, followed by four different phasic aversive stimuli (10 ms 0.7mA footshock, 2 s 1600 lumens bright light presentation and 1 s acoustic 115 dB siren) randomly interleaved with 30 s interval, followed by a 15 min baseline recording period, and then either 10 mg/kg LiCl or saline, i.p. or 6 min restraint stress. For VTA recordings, rats only completed the sessions with reward and phasic aversive stimuli. For Pavlovian conditioning paradigm, once rats achieved 85% accuracy in reward trials, they were trained to respond to an 8 kHz tone (75 dB) lasting for 2 s followed by a mild footshock (0.7mA). During testing, rats again placed on mild food deprivation, and recordings obtained in sessions consisting of 150 mixture of reward trials, neutral trials, and shock trials randomly selected. Rats were recorded for one or two session per day, and electrodes advanced 80–160 µm at the end of each session. Neurons with significant reductions in baseline firing rates across sessions were excluded from the study, as this is indicative of drifting of recording wires between sessions.

## In vivo Ca$^{2+}$ imaging

Wild type mice were placed into operant chambers (Med Associates) after being food deprived to 85% their original body weight. Mice were trained with 1 kHz auditory tones (70 dB, 2 s) followed immediately by sucrose pellet delivery. During training, tones came to elicit approach to the food tray. Once mice reached criterion (approach responses within 3 s on >85% of trials), ad libitum feeding was restored, and mice received injections of AAV2-hSyn-FLEX-CaMP6f virus (UNC Vector Core) into the RMTg (AP: −4; ML: 1.2, DV: −4 from dura, 10-degree angle) and CAV2-Cre (Montpellier Vector Core) into the VTA (AP: −5.1 mm; ML: 2.5 mm; DV: −7.8 mm from dura, 10-degree angle) or DRN (AP: −3.7 mm; ML: −0.6 mm; DV: 3 mm from dura, 10-degree angle). After 3 weeks' recovery, we implanted a gradient index (GRIN) lens (outer diameter 0.5 mm, length 6.0 mm) (Inscopix) with the tip of the lens placed 0.2–0.3 mm dorsal to the viral injection site. Four to 6 weeks after lens implantation, a baseplate was implanted. Before each recording session, the mouse was briefly anesthetized with isoflurane and a miniature camera attached to the baseplate with a setscrew. Each animal was then put back into its home cage for 20 min to recover. We recorded from each animal for two sessions per day: reward session and shock session. In reward sessions, the mice received 30 2 s auditory cues paired with sucrose pellet deliveries. The interval between trials was 30 s. In shock sessions, mice received 30 0.2mA footshocks at 30 s intervals. The endoscopic camera was turned on for 20 s windows centered at the onset of reward-predictive cue or footshock. Ca$^{2+}$ images were acquired at 20 Hz at LED intensity from 30–70% and gain from 1 to 3.5. Using Mosaic software (Inscopix) we then spatially down-sampled videos to 400 × 384 pixels, and temporally down-

sampled to 5 Hz. Individual trials (for reward/shock sessions) were concatenated and motion corrected. Calcium signals (dF/F) from individual cells were extracted by by CNMF-E, which extracts cellular calcium signals with minimal influence from the background (*Pnevmatikakis et al., 2016*). For reward/shock sessions, time courses were calculated for each 20 s interval corresponding to individual trials, and then averaged together for the session. Data points were normalized to baseline calcium activity, and the data were then smoothed using a 0.6 s moving average.

## Conditioned place test

Two groups of rats were tested for place conditioning after recovery from surgery. One group received bilateral RMTg lesions and the other group served as sham controls. The lesion was made prior to behavioral training, and lesion sizes were verified by NeuN staining. Both groups were exposed to intraperitoneal lithium chloride (150 mg/kg, Sigma-Aldrich, 7 lesions and six shams), bright light (1600 lumens, 2 s pulse with 2 s duration, 7 lesions and eight shams) and siren (115 dB, 1 s pulse with 2 s duration, 7 lesions and six shams). We used i.p. saline injection, dim house light and 75 dB noise as control treatments. The conditioned place test experiments were performed using a three-chambered apparatus (Med Associates, St Albans, VT) under dim room light. Each animal received only one aversive treatment or control treatment. On the first day of each experiment, rats explored all three chambers freely for 15 min once per day, and an average baseline preference score was determined for each. Over the next 8 days, rats performed one aversive session and one control session each day. Rats were placed into their previously preferred chambers for 15 min in aversive sessions and into the opposite chambers in control sessions. The order of treatments was counterbalanced across rats. On the ninth day, rats again explored all chambers freely without stimulus exposure. The preference score was defined as the number of seconds spent or number of entries into the stimulus-paired chamber minus the number of seconds spent in the unpaired chamber. We also calculated each animal's preference shift, defined as the post-training preference minus the pre-training preference score. The overall activity of each animal was measured as the number of photobeam breaks.

## Statistical analysis of electrophysiological and behavioral data

RMTg and VTA neuron firing rates in response to phasic stimuli were calculated in 50 ms bins and normalized to 1 s baseline before the onset of stimuli. RMTg firing rates in response to sustained stimuli were calculated in 5 min bins and normalized to 15 min baseline. Neurons with large drifting of the microwire electrodes during recordings were excluded from further analysis. Electrophysiology data were first tested for normality, then transformed to ranked forms if data failed tests of normality ($p < 0.05$, D'Agostino-Pearson test). Latency to maximum responses was calculated as the first bin that reached the peak or trough. Burst analysis was performed with NeuroExplorer, with a burst defined by any pair of spikes less than 80 ms apart, and continuing until the last spike is more than 160 ms separated from the following spike.

Significant responses in neural firing were determined by a threshold of $p < 0.05$ for each neuron's firing rate versus baseline (Wilcoxon signed-rank test for phasic stimuli trials in wire recording experiments, paired t-test for Ca$^+$ recording experiments). Calcium recording data and conditioned place aversion data passed tests of normality ($p > 0.05$, D'Agostino-Pearson test), and were analyzed using parametric tests. Post hoc tests after one-way and two-way ANOVA were Holm-Sidak and Bonferroni, respectively. Calculations were performed using Matlab (Mathworks) and Prism seven software (Graph Pad).

## Additional information

### Funding

| Funder | Grant reference number | Author |
| --- | --- | --- |
| National Institutes of Health | DA037327 | Thomas C Jhou |
| National Institutes of Health | DA032898 | Thomas C Jhou |

The funders had no role in study design, data collection and interpretation, or the decision to submit the work for publication.

## Author contributions

Hao Li, Conceptualization, Investigation, Methodology, Writing—original draft, Project administration, Writing—review and editing; Dominika Pullmann, Maya Eid, Investigation, Methodology; Jennifer Y Cho, Investigation; Thomas C Jhou, Conceptualization, Supervision, Funding acquisition, Methodology, Writing—review and editing

## Author ORCIDs

Thomas C Jhou (iD) http://orcid.org/0000-0001-8811-0156

## Ethics

Animal experimentation: All procedures were conducted under the National Institutes of Health Guide for the Care and Use of Laboratory Animals, and all protocols were approved by Medical University of South Carolina Institutional Animal Care and Use Committee (protocol #3522).

## Decision letter and Author response

Decision letter https://doi.org/10.7554/eLife.41542.014
Author response https://doi.org/10.7554/eLife.41542.015

## Additional files

### Data availability

All data generated or analysed during this study are included in the manuscript and supporting files.

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
