## [Decision Letter]

Thank you for submitting your article "Generality and opponency of rostromedial tegmental (RMTg) roles in valence processing" for consideration by *eLife*. Your article has been reviewed by three peer reviewers, including Geoffrey Schoenbaum as the Reviewing Editor and Reviewer #1, and the evaluation has been overseen by Laura Colgin as the Senior Editor. The following individuals involved in review of your submission have also agreed to reveal their identity: Mark Ungless (Reviewer #2); Carl Lupica (Reviewer #3).

The reviewers have discussed the reviews with one another and the Reviewing Editor has drafted this decision to help you prepare a revised submission.

Summary:

This is an excellent set of experiments that sheds light on responsiveness of RMTg neurons to a variety of aversive USs, the impact of that activity on downstream neurons in VTA, and the involvement of the RMTg in behaviors motivated by aversion. While the reviewers had a few comments and questions, and a few essential revisions, overall all three reviewers agreed that the work was important, well-designed, and that the results clearly support the authors' main conclusions. The work makes a unique and important contribution to this fast moving field.

Essential revisions:

There were three essential revisions identified; all were relatively minor and did not represent major flaws. If they can be quickly addressed, then it may not be necessary to send the revised manuscript back to all reviewers to re-review given the strong positive evaluation from all three reviewers of the work overall. The first was that the authors clarify whether they think the RMTg output is signaling aversiveness itself versus something more akin to a teaching signal for learning about aversiveness. This can be clarified simply in the Introduction/Discussion or there can be additional data presented to make it clear which the authors are suggesting. The second was that the authors examine the strength of the data in Figure 2L for concluding that there is a reversal of the place preference effects for both LiCL and cocaine in the late phase. This conclusion would be bolstered by the addition of saline controls. The third is the removal or explication of the claim in the Discussion that optotagging is superior to traditional methods for identifying dopamine neurons.

*Reviewer #1:*

In this study the authors assay the responses of RMTg neurons to a reward-paired cue, reward, and several brief and prolonged aversive outcomes. They report that many RMTg neurons show a bidirectional response at short-latency, firing to aversive US's and suppressing firing to the reward-paired cue. They further show using Ca+ imaging that this response was present in RMTg neurons that projected to VTA; neurons projecting to DRN did not show this bidirectional pattern. They further showed that lesioning RMTg impacted firing in VTA neurons, causing them to be less inhibited and more excited by the aversive US's. Lesions also disrupted aversive place conditioning. Overall the work is really fantastic. The goals and ideas are important and interesting. The experiments are well-designed to address the authors' specific question regarding the responsiveness of these neurons to aversive signals and the impact their firing may have downstream, and the results are sure to be of major interest to the field. I only have a couple of questions or objections.

Maybe my only major overarching concern is that I was never clear whether the authors were relating their findings to error signaling or whether they were agnostic on this question. Or trying to make a distinction even. To put it another way: when RMTg neurons fire to an unsignaled shock, do the authors think they signal aversiveness or errors in aversiveness prediction? And do the lesions affect place preference because the shock is not so bad without RMTg? Or because there is no "aversion prediction error" to drive learning to predict the shock? Do the rats freeze or jump or show other unconditioned responses to the shock just like controls? And depending on their answer to these questions, I'd next wonder whether they think RMTg neurons encode anything about what actually happened – is this the significance of the heterogeneity of coding across the US's? Can the authors comment on some of this? Or if not, can they clarify their position and open this as a question for the Discussion? Relatedly – is the reward-paired cue supposed to be rewarding? Were the mice or rats pre-trained? Can you show the behavioral data that they knew it predicted reward? I assume this is the case, since the neurons – RMTg and DA seem to respond to the cue and not the reward, but this is never shown or explained.

*Reviewer #2:*

Li and colleagues present an important series of experiments that greatly expand our understanding of the role of the rostromedial tegmental nucleus (RMTg) in processing aversive stimuli. The manuscript is well-written, the results are presented in clear and logical manner, and they are interpreted in a balanced way. In my view, an important strength of this manuscript is the use of several distinct types of aversive stimuli, which is not common in the field.

On the whole, I am sympathetic towards the methods used to identify putative dopamine neurons and the balanced way this is discussed in the Results and Discussion. However, the authors should note that Root et al., 2018, have reported a small population of vGluT2-expressing VTA neurons that are excited by reward-predicting cues. It seems unlikely to me that all of the pDA neuron in this study are in fact vGluT2/non-dopamine neurons, but it is possible that some are.

It would be helpful to see some examples of the RMTg and VTA electrophysiological recordings.

Introduction, second paragraph. I would specify what type of response was measured i.e., action potential rate).

Introduction, third paragraph. I would include GABA interneurons in this list.

Subsection “RMTg neurons are activated by diverse phasic aversive stimuli”, first paragraph. It would be helpful here to expand on the different aspects of cocaine. Strictly speaking the stimulus was 'cocaine' rather than the 'aversive effect of cocaine'.

Subsection “Excitotoxic RMTg lesions abolished VTA inhibitions by aversive stimuli”, third paragraph, "affected" should be "affect".

Subsection “Broad encoding of valence in RMTg contributes to VTA valence encoding”, first paragraph, "Expect" should be "Except".

Figure 2—figure supplement 1 legend. In C and D 'RMTg' should be 'VTA'.

*Reviewer #3:*

In this extensive series of experiments, the investigators explore RMTg neuron coding of a wide-range of negative and positive environmental stimuli and the contributions of these neurons to place-preference and -aversion in rodents. As RMTg neurons are known to be inhibitory GABAergic cells providing strong and extensive input to ventral midbrain dopamine neurons involved in processing reward and aversion, a more complete understanding of their encoding of a broad range of environmental stimuli is central to a more comprehensive view of their roles in behavior. Using multi-wire recording of RMTg neuron activity, the authors report that rewarding stimuli largely inhibited RMTg cells, whereas presentation of aversive stimuli was associated with increased excitation. Moreover, using in vivo calcium imaging in RMTg neurons projecting to either the ventral tegmental area (VTA) or the dorsal raphe nucleus (DRN), the authors show a preferential projection of stimuli-responsive neurons to the VTA. The authors also show that lesions of RMTg neurons prevent conditioned-place aversion to aversive stimuli and that the temporal dynamics of RMTg neuron biphasic responses to sustained aversive stimuli (i.e. LiCl, cocaine, or restraint) are associated with place-aversion or place-conditioning. This latter series of experiments provides strong evidence that RMTg neurons demonstrating biphasic responses may underlie opponent behavioral processes that have long been described in the learning and behavior literature. Moreover, these studies provide one of the few examples supporting a neuronal mechanism associated with negative reinforcement.

My general impression of the work is that it makes a strong contribution to our understanding of the roles that RMTg neurons play in processing environmental rewarding and aversive stimuli and clarifies their contribution to the regulation of dopamine neuron activity in these behavioral processes. The study is comprised of an ambitious series of experiments with a strong singular focus on defining RMTg responses to a broad range of stimuli and to elucidating the contribution of these responses to shaping reward and aversive behavior. Each of the experiments is well-constructed and executed, and the conclusions drawn from these studies are strongly supported by the data.

---

## [Author Response]

Reviewer #1:

[…] I only have a couple of questions or objections.Maybe my only major overarching concern is that I was never clear whether the authors were relating their findings to error signaling or whether they were agnostic on this question. Or trying to make a distinction even. To put it another way: when RMTg neurons fire to an unsignaled shock, do the authors think they signal aversiveness or errors in aversiveness prediction?

This is a very interesting question, and it is our observation that, unlike DA neurons, RMTg neurons encode both aversiveness and prediction errors, i.e. they respond with excitation to shocks even if they are fully predicted by prior cues, while also showing an enhanced excitation if the shock is unpredicted. The current paper only uses unpredicted shocks, and hence is not able to address this issue, but one other publication we are preparing examines this issue in depth.

And do the lesions affect place preference because the shock is not so bad without RMTg? Or because there is no "aversion prediction error" to drive learning to predict the shock? Do the rats freeze or jump or show other unconditioned responses to the shock just like controls?

Again, this is a very interesting question that falls outside the scope of the current study, but data from other labs, as well as our own published and unpublished findings, show that freezing is nearly abolished in RMTg-lesioned rats, while escape behaviors (e.g. running away from a shock) appear largely intact. Hence, it appears the RMTg clearly contributes to some types of aversive responses (punishment and fear), but not others (escape). We now discuss this at greater length near the end of the Discussion.

And depending on their answer to these questions, I'd next wonder whether they think RMTg neurons encode anything about what actually happened – is this the significance of the heterogeneity of coding across the US's? Can the authors comment on some of this?

Because the responses to the various US’s all seemed to correlate with each other, we lean towards the assumption that the RMTg does not convey much information about the specific identity of the aversive stimulus. The RMTg and VTA are of similar size, i.e. small, while the numbers of possible rewarding or aversive stimuli an animal might encounter are vast. So it would be hard to imagine how these small regions, with relatively few neurons, would convey information about specific stimulus identities. However, we did not do any experiments to explicitly test this. Also, if RMTg heterogeneity were to encode stimulus identity (rather than simply valence), one would expect to see DA neuron heterogeneity reflecting a similar encoding, and we are not aware of such a precedent (which of course, doesn’t mean that such a relationship wouldn’t exist).

Or if not, can they clarify their position and open this as a question for the Discussion? Relatedly – is the reward-paired cue supposed to be rewarding? Were the mice or rats pre-trained? Can you show the behavioral data that they knew it predicted reward? I assume this is the case, since the neurons – RMTg and DA seem to respond to the cue and not the reward, but this is never shown or explained.

Rats were trained with Pavlovian conditioning before recordings, and are required to reach >80% discrimination between the rewarded versus neutral cues. Accurate discrimination consists of nosepoking into the food port within two seconds after onset of the reward-paired but not the neutral cues, as now shown in Figure 1B. Since the reward is not delivered until 2 seconds after cue onset, any nosepoke made during this time is taken as evidence of reward expectation, while withholding of nosepokes during this period is taken as evidence that the animal does not expect reward.

Regarding the question of exactly why the RMTg lesions affect place preference, this is a difficult question to address for the same reasons as noted above. We address some of the complexities in our expanded Discussion, which specifically talks about the fact that RMTg lesions greatly impair the aversion-induced bias in entries, but not time spent in, the various chambers.

Reviewer #2:

[…] On the whole, I am sympathetic towards the methods used to identify putative dopamine neurons and the balanced way this is discussed in the Results and Discussion. However, the authors should note that Root et al., 2018, have reported a small population of vGluT2-expressing VTA neurons that are excited by reward-predicting cues. It seems unlikely to me that all of the pDA neuron in this study are in fact vGluT2/non-dopamine neurons, but it is possible that some are.

This has been noted in the Discussion.

It would be helpful to see some examples of the RMTg and VTA electrophysiological recordings.

Examples of RMTg recording have been added as Figure 2B, and examples of VTA recording were included in the Figure 4.

Introduction, second paragraph. I would specify what type of response was measured i.e., action potential rate).

Response type has been specified as firing rates.

Introduction, third paragraph. I would include GABA interneurons in this list.

Fixed.

Subsection “RMTg neurons are activated by diverse phasic aversive stimuli”, first paragraph. It would be helpful here to expand on the different aspects of cocaine. Strictly speaking the stimulus was 'cocaine' rather than the 'aversive effect of cocaine'.

Fixed.

Subsection “Excitotoxic RMTg lesions abolished VTA inhibitions by aversive stimuli”, third paragraph, "affected" should be "affect".

Fixed.

Subsection “Broad encoding of valence in RMTg contributes to VTA valence encoding”, first paragraph, "Expect" should be "Except".

Fixed.

Figure 2—figure supplement 1 legend. In C and D 'RMTg' should be 'VTA'.

Fixed.